# Adaptation by copy number variation increases insecticide resistance in the fall armyworm

Sylvie Gimenez[1,8], Heba Abdelgaffar[2,8], Gaelle Le Goff[3], Frédérique Hilliou[3], Carlos A. Blanco[4], Sabine Hänniger[5], Anthony Bretaudeau[6,7], Fabrice Legeai[6,7], Nicolas Nègre[1], Juan Luis Jurat-Fuentes[2], Emmanuelle d'Alençon[1] & Kiwoong Nam[1✉]

Understanding the genetic basis of insecticide resistance is a key topic in agricultural ecology. The adaptive evolution of multi-copy detoxification genes has been interpreted as a cause of insecticide resistance, yet the same pattern can also be generated by the adaptation to host-plant defense toxins. In this study, we tested in the fall armyworm, *Spodoptera frugiperda* (Lepidoptera: Noctuidae), if adaptation by copy number variation caused insecticide resistance in two geographically distinct populations with different levels of resistance and the two host-plant strains. We observed a significant allelic differentiation of genomic copy number variations between the two geographic populations, but not between host-plant strains. A locus with positively selected copy number variation included a CYP gene cluster. Toxicological tests supported a central role for CYP enzymes in deltamethrin resistance. Our results indicate that copy number variation of detoxification genes might be responsible for insecticide resistance in fall armyworm and that evolutionary forces causing insecticide resistance could be independent of host-plant adaptation.

[1] DGIMI, Univ of Montpellier, INRA, Place Eugène Bataillon, 34095 Montpellier, France. [2] Department of Entomology and Plant Pathology, University of Tennessee, 370 Plant Biotechnology Building, 2505 E J. Chapman Dr, Knoxville, TN 37996, USA. [3] Université Côte d'Azur, INRAE, CNRS, ISA, 400 Route des Chappes, 06903 Sophia Antipolis, France. [4] United States Department of Agriculture, Animal and Plant Health Inspection Service, 4700 River Rd, Riverdale 20737 MD, USA. [5] Max Planck Institute for Chemical Ecology, Hans-Knoell-Straße 8, 07745 Jena, Germany. [6] IGEPP, INRAE, Institut Agro, Univ Rennes, Campus de Beaulieu, 263 Avenue Général Leclerc, 35042 Rennes, France. [7] GenOuest Core Facility, Univ Rennes, Inria, CNRS, IRISA, Campus de Beaulieu, 263 Avenue Général Leclerc, 35042 Rennes, France. [8] These authors contributed equally: Sylvie Gimenez, Heba Abdelgaffar. ✉email: ki-woong.nam@inrae.fr

The emergence of insecticide resistance is one of the biggest challenges in pest control, costing billions of dollars every year in the US alone[1]. The identification of genes responsible for insecticide resistance has been one of the main topics in agricultural ecology because this knowledge can be used to design effective ways of controlling pests[2]. Well-known detoxification genes include cytochrome P450s (CYPs)[3], esterases[4], Glutathione S-transferases (GSTs)[3], UDP glucuronosyltransferases (UGTs)[5], and oxidative stress genes[6]. Copy number variation (CNV) of these detoxification genes and associated positive selection have been reported from several insect pest species, including mosquitoes (*Anopheles gambiae*)[7,8], tobacco cutworms (*Spodoptera litura*)[9], and fall armyworms (*S. frugiperda*)[10]. These genetic mechanisms are assumed to be involved in insecticide resistance[11].

However, positive selection on detoxification genes is not necessarily a cause of insecticide resistance. Insects produce detoxification proteins to overcome plant defense toxins. Thus, the adaptive CNV of detoxification genes can be a consequence of the evolutionary arms race between insects and plants[12]. As the evolutionary history of host-plant adaptation is much longer than that involving human insecticide application[1], the vast majority of evolutionary genetic footprints of detoxification genes might be generated independently from insecticides. Therefore, the association between the CNV of detoxification genes and insecticide resistance remains elusive.

The fall armyworm (*Spodoptera frugiperda*, Lepidoptera: Noctuidae) is one of the most damaging pest insects of different crop plants due to its extreme polyphagy and strong migratory behavior. While native to the subtropical regions of the Americas, this species invaded sub-Saharan Africa in 2016[13] and then globally spread to India[14], South East Asia, East Asia, Egypt, and, more recently, Australia (https://www.cabi.org/isc/fallarmyworm). Interestingly, this species consists of two sympatric strains, the corn (sfC) and rice (sfR) strains[15–17]. Although host choice is not absolute, sfC larvae prefer feeding on tall grasses like corn or sorghum, while sfR larvae prefer smaller grasses like rice or pasture grasses. The differentiated host-plant range implies different adaptation processes to host-plants, probably involving differentiated detoxification processes[18]. In Puerto Rico, fall armyworms show a dramatically increased level of field-evolved resistance to a wide range of insecticides compared with populations in the US mainland (e.g., Monsanto strain)[19] and Mexico[20]. The reason for this increase is not well known, but one of the possibilities is strong selective pressure by frequent sprays of insecticides for corn seed production[21].

Consequently, the fall armyworm population from Puerto Rico is an ideal model species to distinguish the adaptive role of CNVs between host-plant adaptation and insecticide resistance. If positive selection specific to Puerto Rico's population is observed from the CNVs of detoxification genes while these CNVs are not specific to strains, then the adaptive role of CNVs in insecticide resistance would be supported. Alternatively, if all identified positive selection by CNVs is specific to the host-plant strains, then the CNVs would contribute predominantly to host-plant adaptation. To test this hypothesis, we analyzed the resequencing data from fall armyworms in Puerto Rico and Mississippi, including sfC and sfR for each population, together with a new high-quality genome assembly with chromosome-sized scaffolds. In addition, we used toxicological bioassays to test the participation of cytochrome P450 enzymes in resistance to deltamethrin in *S. frugiperda* from Puerto Rico compared to Mississippi.

## Results

**Reference genome and resequencing data.** We generated a new sfC genome assembly using PacBio reads, displaying 27.5X coverage. The resulting assemblies are 384 Mb in size, and N50 is 900kb. Super-scaffolding of this assembly was performed using Hi-C[22] data and HiRise software[23]. The final assembly size is 384.46 Mb, which is close to the expected genome size estimated from flow cytometry (396 ± 3 Mb)[10]. The number of sequences is 125, and N50 is 13.15 Mb. The number of the longest superscaffolds explaining 90% of genome assembly (L90) is 27. The scaffold lengths show a bimodal distribution, and a mode corresponding to the longest sequences (>5 Mb) contains 31 scaffolds (Supplementary fig. 1). As the fall armyworm has 31 chromosomes in the haploid genome, this pattern demonstrates chromosome-sized scaffolds in the assembly. Recently, Zhang et al. also generated a high-quality chromosome-sized assembly from an invasive African population[24]. We believe that these two chromosome-sized assemblies are complementary as the assemblies generated in this study and by Zhang et al. represent native and invasive populations, respectively.

Recently, Liu et al. also published new reference genome assemblies from invasive fall armyworms[25]. These assemblies are substantially larger than the expected genome size (530.77–542.42 Mb), raising the possibility of misassemblies. As Liu et al. generated the genome assemblies from a natural population, a potentially high level of heterozygosity might inflate the assembly size. To evaluate the correctness of the assemblies, we performed BUSCO (Benchmarking Universal Single-Copy Ortholog) analysis[26]. Our new assembly has a higher proportion of complete BUSCO genes (Table 1) and a lower proportion of duplicated BUSCO genes (28/1658 = 1.69%) than the assemblies by Liu et al.[25] (134/1658 = 8.08% for male assembly, 97/1658 = 5.86% for female assembly). From this BUSCO analysis, we concluded that our assembly has higher accuracy and better contiguity, because of a much smaller number of contigs and a

**Table 1 Summary statistics for the reference *S. frugiperda* genome assembly and the result of BUSCO analysis from the assembly used in this study and in Liu et al.[25].**

| Assembly statistics | Current assembly | Liu et al., male | Liu et al., female |
|---|---|---|---|
| Assembly size (bp) | 384,455,365 | 543,659,128 | 531,931,622 |
| Number of sequences | 125 | 21,840 | 27,258 |
| N50 (bp) | 13,151,234 | 14,162,803 | 13,967,093 |
| L50 | 13 | 16 | 17 |
| N90 (bp) | 8,473,354 | 6440 | 5122 |
| L90 | 27 | 3030 | 5122 |
| Length of gaps (bp) | 346,864 | 37,953,553 | 35,713,062 |
| Complete and single-copy BUSCOs | 1573 | 1442 | 1480 |
| Complete and duplicated BUSCOs | 28 | 134 | 97 |
| Fragmented BUSCOs | 20 | 45 | 48 |
| Missing BUSCOs | 37 | 37 | 33 |

much smaller L90 value than the assemblies from Liu et al.[25] (Table 1).

Resequencing data were obtained from *S. frugiperda* populations from Mississippi (MS) and Puerto Rico (PR) using Illumina sequencing technology. We used mitochondrial genomes[27] to identify host strains for each individual (Supplementary fig. 2). Resequencing data of MS contained nine sfC and eight sfR individuals, while data from PR contained 11 sfC and four sfR individuals. We then performed mapping of the resequencing data against the reference genome assembly. SNV (single nucleotide variants) and CNV were identified using GATK[28] and CNVcaller[29], respectively (Supplementary fig. 3). For the CNVs, we used only those showing minor allele frequencies greater than 0.1 in order to reduce false positives. We identified 41,645 CNVs with an average CNV unit size of 1501.14 bp (800 bp–151 kb). We identified 16,341,783 SNVs, after strict filtering.

**CNV according to geographical differentiation**. We performed principal component analysis (PCA) to infer the genetic relationship among individuals from each of the CNVs and SNVs. The result from CNVs showed a clear grouping according to geographical population, whereas the grouping according to the host strain was not observed (Fig. 1). SNVs also showed a weak pattern of grouping according to geographical population, but not according to host strain. The PCA result shows that individuals from PR had a much smaller variation of CNV than individuals from MS, while SNVs did not show such a pattern. This result is in line with prevalent natural selection by CNVs specific to PR, which is not observed from SNVs.

$F_{ST}$ values calculated from CNVs and SNVs between PR and MS were 0.142 and 0.0163, respectively, and both $F_{ST}$ values are significantly greater than the expectation based on random grouping ($p < 0.001$ for both CNV and SNV). This result implies that both CNVs and SNVs show significant genetic differentiation between PR and MS, with a much greater extent for CNV, in line with the PCA results (Fig. 1). $F_{ST}$ calculated from CNV between strains was not significantly greater than the expectation based on random grouping ($F_{ST} = 0.0019$, $p = 0.262$; one-sided randomization test), while $F_{ST}$ calculated from SNV is very low but significantly higher than the expectation by random grouping ($F_{ST} = 0.0093$, $p < 0.001$; one-sided randomization test). These results support that the genomic distribution of CNVs has been (re)shaped by evolutionary forces that are specific to geographic population(s), but not specific to host-plants.

Importantly, $F_{ST}$ values between sfC and sfR may not represent the level of genetic differentiation between the strains if F1 hybrids exist in the samples used in this study. All the samples used in this study were females, which have ZW sex chromosomes (in Lepidoptera males are the homogametic sex with ZZ chromosomes). The Z chromosome in females is paternally inherited, whereas the mitochondrial genomes are maternally inherited. Therefore, female Z chromosomes and mitochondrial genomes in F1 hybrids are inherited from different host strains. Since we used mitochondrial markers to identify host strain, if a substantial proportion of these samples were hybrids, Z chromosomes would have lower $F_{ST}$ values between strains than autosomal genomes (see Supplementary fig. 4 for more details). We performed blast searches of Z-linked TPI genes[30] to identify Z chromosomes in the assembly, and we observed only a single blast-hit, at scaffold 66. This scaffold is 21,694,391 bp in length. This Z chromosome had higher $F_{ST}$ than autosomes ($F_{ST}$ for Z chromosome and autosome is 0.024 and 0.0088, respectively, $p = 0.0085$; one-sided bootstrapping test). Therefore, it is unlikely that F1 hybrids affected the results with an observable extent.

**CNVs under positive selection**. We inferred geographic population-specific positive selection from the loci with nearly complete allelic differentiation of CNVs between PR and MS ($F_{ST} > 0.8$). In total, we identified seven positively selected loci (Fig. 2a). Among these loci, the unit of the CNV at scaffold 17 is a gene cluster composed of 12 CYP9A genes and two alcohol dehydrogenase (ADH) genes (Fig. 2b). This gene cluster was also observed in our previous study based on BACs from sfC[31]. In PR, all 30 alleles have two copies of this unit, while 28 and 6 alleles in the MS had one and two copies, respectively. The presence of a single haplotype of CNVs in PR and of two haplotypes of CNVs in MS implies the existence of positive selection specific to PR because positive selection fixes only a single haplotype in a population. Overexpression of CYP9A genes upon treatment with insecticides was reported in the fall armyworm[32], beet armyworm (*S. exigua*)[33], tobacco cutworm (*S. litura*)[34], and smaller tea tortrix (*Adoxophyes honmai*)[35], supporting the association between CYP9A and insecticide resistance. CYP9A12 and CYP9A14 were overexpressed in a pyrethroid-resistant strain of *Helicoverpa armigera*[36], and functional expression of these two CYP genes demonstrated their ability to metabolize pyrethroids[37]. In *H. armigera*, ADH5 binds a promoter of CYP6B6 in the response of xenobiotic 2-tridecanone[38]. Deciphering the functions of CYP and ADH genes from scaffold 17 will contribute to understanding insecticide resistance in PR.

The almost completely differentiated CNVs also include a sequence at scaffold 24 (Fig. 2a). In PR, the number of copies was always greater than three, while 88.2% of alleles in MS had one or two copies. Because multiple haplotypes exist in PR and MS, it is unclear whether positive selection occurs specifically in PR because positive selection may also occur in MS in the way of

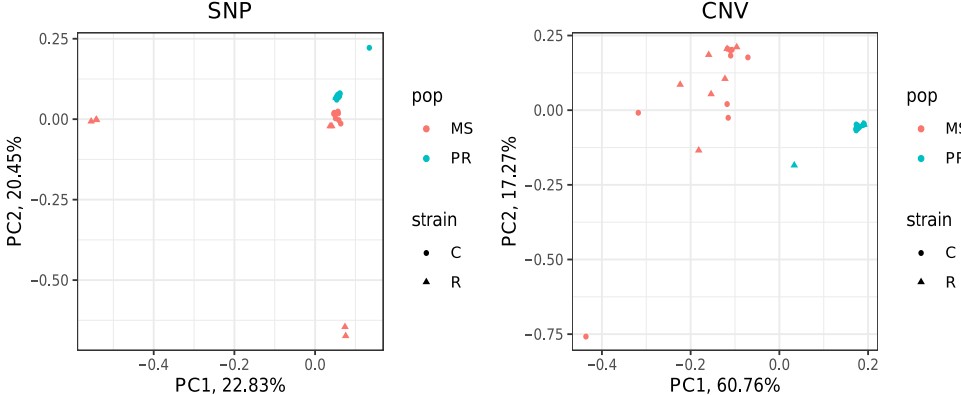

**Fig. 1 Principal component analysis.** The left and right panels show the results from SNP and CNV, respectively. MS and PR represent samples from Mississippi and Puerto Rico, respectively. The C and R represent the corn (sfC) and rice (sfR) host strains, respectively.

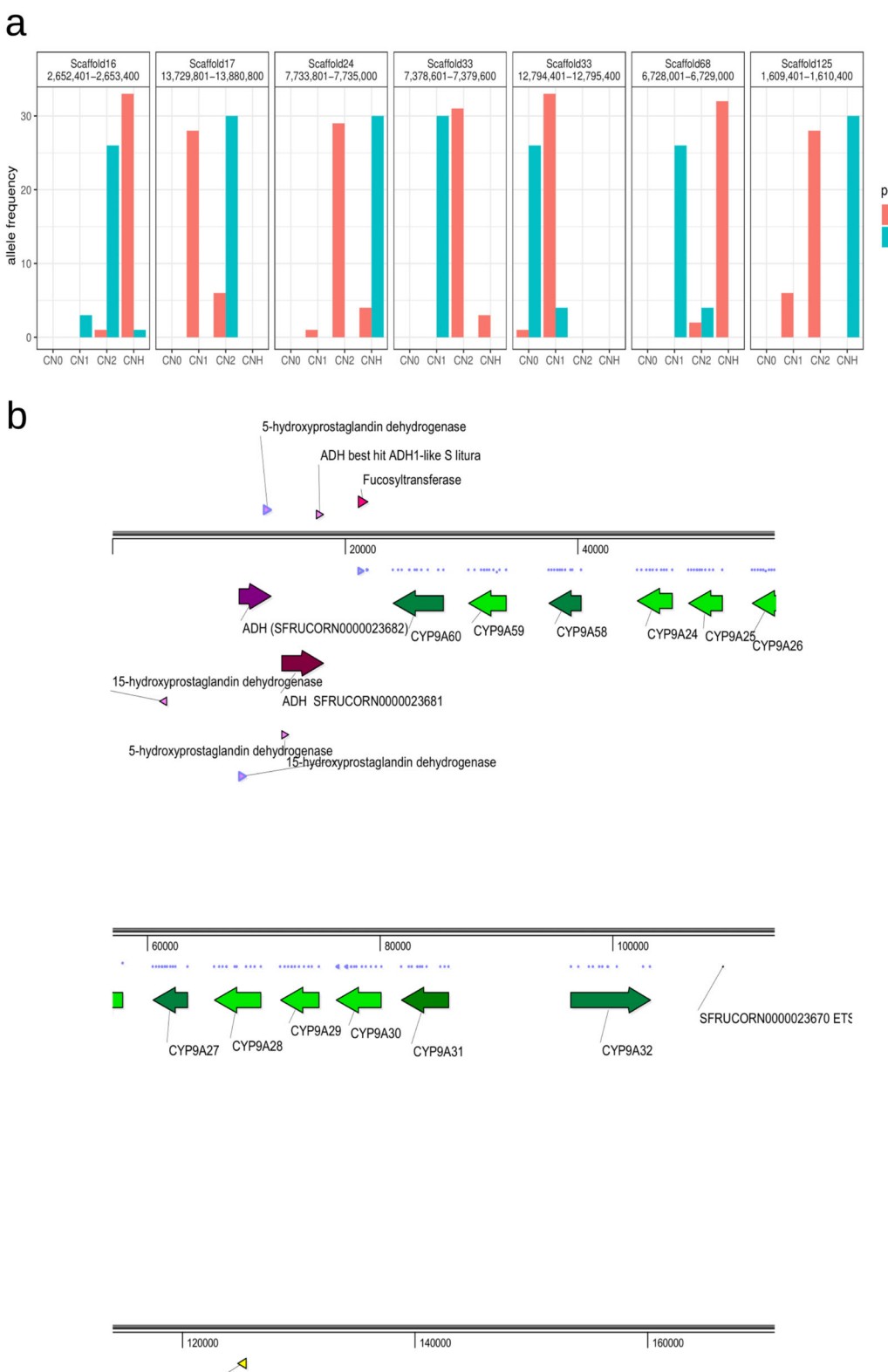

**Fig. 2 Positively selected CNV. a** Allele frequency of positively selected CNV. CN0, CN1, CN2, and CNH represent copy number equal to zero, one, two, and greater than two, respectively. **b** The genes within the positively selected loci at the loci in scaffold 17. The purple and green colors represent ADH genes and CYP genes, respectively. The arrows indicate the direction of transcription.

reducing copy numbers. This CNV contains a chitin deacetylase gene, which is widely used as a target of insecticides[39], suggesting the possibility that CNV of this gene increased resistance in PR. Chitin deacetylase is also associated with the response to *Bacillus thuringiensis* (Bt) in *H. armigera*[40]. Populations of *S. frugiperda* in PR developed resistance against transgenic corn producing Bt toxins[19,41,42], and this resistance is linked to mutations in an ATP binding cassette C2 gene[43,44]. However, we speculate that the observed chitin deacetylase CNV may contribute to Bt-resistance in *S. frugiperda* from PR as well.

Scaffold 68 also contains almost completely differentiated CNV (Fig. 2a), containing a gustatory receptor gene. The copy number varied both in PR and MS. The CNV of this gene is reported to be associated with the interaction with host-plants and might be related to adaptation to feed on these hosts[10,34]. However, the allelic differentiation between sfC and sfR was not observed at this loci. We did not identify any other protein-coding genes of known function from the remaining positively selected CNVs.

**Toxicological bioassays**. We performed toxicological bioassays to experimentally test the putative role of the detected detoxification enzyme gene CNV in PR versus MS samples for susceptibility to pesticides using a pyrethroid (deltamethrin) in larvae from *S. frugiperda* strains originated from Mississippi (Benzon) and Puerto Rico (456LSD4, which was seeded from the PR populations used in this study). Results from these bioassays detected a 6.5-fold resistance to deltamethrin (95% confidence limits 4.7–9.1) when comparing the lethal concentration killing 50% of the larvae ($LC_{50}$) between the 456LSD4 and the Benzon strains (Table 2).

Piperonyl butoxide (PBO) is an inhibitor of CYP enzymes commonly used in bioassays to determine the role of these detoxification enzymes in resistance to pesticides[45,46]. Including a constant (non-lethal) amount of PBO in bioassays with deltamethrin resulted in the synergism of activity against 456LSD4 larvae (Table 2), so that $LC_{50}$ values were not significantly different (overlapping 95% confidence intervals), from Benzon.

We also confirmed the role of CYP detoxification in insecticide resistance in 456LSD4 from the slopes and the intercepts of probit linear regression lines between the log-transformed dose of deltamethrin and the proportion of dead larvae[47]. The 456LSD4 strain exhibited a lower slope than the Benzon strain ($p = 6.16 \times 10^{-7}$; log-likelihood ratio test, degree of freedom = 1) (Table 2). As 456LSD4 has a lower intercept than Benzon (−0.558 versus 0.644), this result again supports that 456LSD4 strain has resistance compared to the Benzon strain. When PBO was added, the difference in slopes is not statistically significant ($p = 0.215$). Interestingly, in this case, the log-likelihood ratio test rejected ($p = 0.162$; degree of freedom = 2) a null hypothesis of equality (i.e., two linear regression lines having the same slope and intercepts), implying that resistance in 456LSD4 was eliminated by treatment with PBO. This result further supports the hypothesis that CYP genes are directly involved in resistance to insecticides in *S. frugiperda* populations from Puerto Rico compared to

populations from Mississippi. Moreover, these results potentially include the identified CYP9A gene CNVs targeted by PR-specific positive selection (see section "CNVs under positive selection").

**Detoxification genes with CNV**. We tested if genes with CNVs were overrepresented in the list of well-known detoxification genes, such as CYP, esterase, Glutathione S-transferase (GST), UDP glucuronosyltransferases (UGT), and oxidative stress genes[10]. We observed that genes with CNV were significantly overrepresented in all the lists (FDR corrected $p < 0.10$) with the exception of CYP (FDR corrected $p$-value = 0.211) (Fig. 3). This result supports the association between detoxification and genomic CNVs.

We also performed gene ontology analysis to test the overrepresentation of other gene categories with CNVs. In total, 31 gene ontology terms were overrepresented in the list of genes with CNVs (Table 3). These terms included anatomical structure development, developmental process, female gamete generation, and drug binding. This result shows that CNV might be functionally associated with other phenotypes in addition to detoxification.

**Beneficial effects of CNV**. In investigating the overall fitness effects of the observed CNVs, we hypothesized that, if the CNVs generate beneficial effects, they might be under the process of positive selection while yet to be fixed in a population. Alternatively, if the vast majority of CNVs have neutral or slightly deleterious effects, then the CNVs may evolve in an effectively neutral way. To test these hypotheses, we first compared the strength of selection between CNVs and SNVs with an assumption that exons are under stronger selective pressure than non-exonic sequences. The proportion of CNVs containing exonic sequences was 28.14% (11,720/41,645). This proportion is higher than that of SNVs (11.31%, 1,847,439/16,341,783, $p < 2.2 \times 10^{-16}$; two-sided Fisher's exact test). The same trend was observed in mosquitoes, in which CNVs are overrepresented in gene-containing regions[8].

This result can be interpreted by one of the following two possibilities. The first explanation is that the CNVs are experiencing stronger positive selection than SNVs because CNVs have a higher proportion of beneficial variants than SNV. The second explanation is that CNVs are under weaker purifying selection than SNVs because CNVs have weaker deleterious effects than SNVs. According to population genetics theory, when purifying selection is weak, the proportion of rare alleles is increased because slightly deleterious variants still remain unpurged as rare variants in a population. To test the second possibility, we compared the proportion of singleton variants between CNVs and SNVs using the unfiltered CNV dataset. We observed that CNVs have a much lower proportion of singleton polymorphisms than SNVs (0.29% for CNV, 94.0% for SNV, $p < 2.2 \times 10^{-16}$), suggesting that weaker purifying selection on CNVs is not supported. Instead, stronger positive selection on CNV is a more likely scenario. The folded site frequency spectrum is very different between CNV and SNV

**Table 2 Mortality parameters in *S. frugiperda* from Mississippi (Benzon) and Puerto Rico (456LSD4) exposed to deltamethrin alone or in the presence of piperonyl butoxide (PBO).**

| Strain | Origin | Treatment | LC50 | 95% CI | Slope (SE) |
|---|---|---|---|---|---|
| Benzon | Mississippi | deltamethrin | 0.24 | 0.19–0.31 | 2.02 (0.17) |
| 456LSD4 | Puerto Rico | deltamethrin | 1.57 | 0.94–3.08 | 1.12 (0.08) |
| | | deltamethrin+ PBO | 0.29 | 0.22–0.38 | 2.39 (0.25) |

Units are picograms of active ingredient per centimeter square of diet surface. CI = Confidence intervals (95%), SE = standard error.

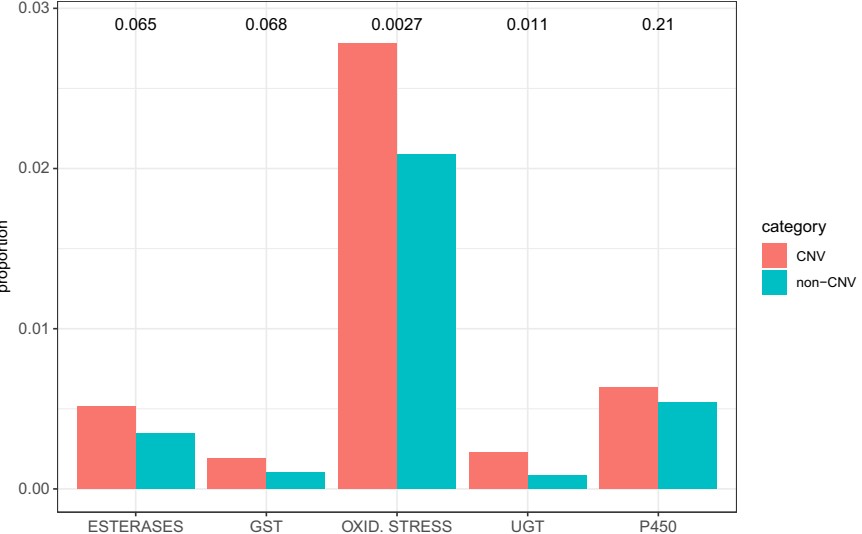

**Fig. 3 Testing overrepresentation of detoxification genes with CNV.** The bars indicate the proportion of detoxification genes in the genes with CNV or genes without CNV. The numbers above the bars indicate FDR-corrected *p* values.

**Table 3 The overrepresented gene ontology terms in the genes with CNV.**

| Category | Description | FDR-adjusted *p*-value |
|---|---|---|
| Biological process | Developmental process | 0.00184 |
| | Localization | 0.01494 |
| | Anatomical structure development | 0.04502 |
| | Establishment or maintenance of cell polarity | 0.06076 |
| | Multicellular organismal process | 0.06168 |
| | Multicellular organism development | 0.06377 |
| | Anatomical structure morphogenesis | 0.06938 |
| | Female gamete generation | 0.06938 |
| Cellular component | Chromosomal part | 0.05904 |
| Molecular function | Adenyl ribonucleotide binding | 0.00003 |
| | Adenyl nucleotide binding | 0.00003 |
| | ATP binding | 0.00003 |
| | Drug binding | 0.00006 |
| | Purine ribonucleotide binding | 0.00010 |
| | Purine nucleotide binding | 0.00010 |
| | Purine ribonucleoside triphosphate binding | 0.00010 |
| | Ribonucleotide binding | 0.00012 |
| | Carbohydrate derivative binding | 0.00021 |
| | Anion binding | 0.00022 |
| | ATPase activity | 0.00155 |
| | Small molecule binding | 0.00189 |
| | Nucleoside phosphate binding | 0.00264 |
| | Nucleotide binding | 0.00264 |
| | Binding | 0.00283 |
| | ATPase activity, coupled | 0.00506 |
| | Ion binding | 0.01027 |
| | Protein binding | 0.01036 |
| | Hydrolase activity, acting on acid anhydrides | 0.06928 |
| | Hydrolase activity, acting on acid anhydrides, in phosphorus-containing anhydrides | 0.06928 |
| | Nucleoside-triphosphatase activity | 0.08469 |
| | Pyrophosphatase activity | 0.08535 |
| | Microtubule motor activity | 0.08535 |

(Fig. 4a). Neutral or slightly deleterious effects are not likely to cause the observed pattern of folded site frequency spectrum of CNV, because, in this case, the proportion of singleton variants is expected to be highest among all site frequencies even in the presence of a bottleneck[48]. Instead, the process of increasing derived allele frequency is a more likely explanation of this folded site frequency spectrum. The ancestry coefficient analysis showed that PR has a smaller number of ancestral CNV haplotypes than MS, supporting stronger positive selection in PR (Fig. 4b).

## Discussion

Results in this study support that the adaptive evolution of detoxification genes by CNV increased the level of insecticide resistance in fall armyworms. The analysis of the genomic CNVs between geographical populations with different levels of insecticide resistance and between strains with different host-plants enabled us to distinguish the roles played by CNV between insecticide resistance and host-plant adaptation. We observed significant allelic differentiation of CNV between MS and PR, but not between host-plant strains. The loci with almost complete allelic differentiation between the geographic populations include a gene cluster containing CYP genes, which are one of the key players of the detoxification process, implying that this CNV potentially plays a crucial role in the increased level of insecticide resistance. The distribution of CNV across the genome is in line with beneficial effects.

Results from bioassays with deltamethrin in fall armyworm larvae further support the role of detoxification enzymes (CYP) in resistance to deltamethrin in a strain from Puerto Rico compared to a reference strain from Mississippi. Deltamethrin and PBO were selected for these bioassays because CYP enzymes are involved in *S. frugiperda* resistance to deltamethrin[49] and are inhibited by PBO. Since neither of the used strains was exposed to chemical pesticides during rearing, these results are in agreement with previous reports and evidence higher resistance to pesticides in *S. frugiperda* populations from Puerto Rico compared to Mississippi[20]. The synergism observed with PBO, which eliminated resistance to deltamethrin in 456LSD4, supports that resistance to deltamethrin in 456LSD4 is mediated by CYP detoxification. This observation is in line with the CNV of CYP genes being responsible for increased insecticide resistance in PR, as shown in *Drosophila melanogaster* that the duplication of

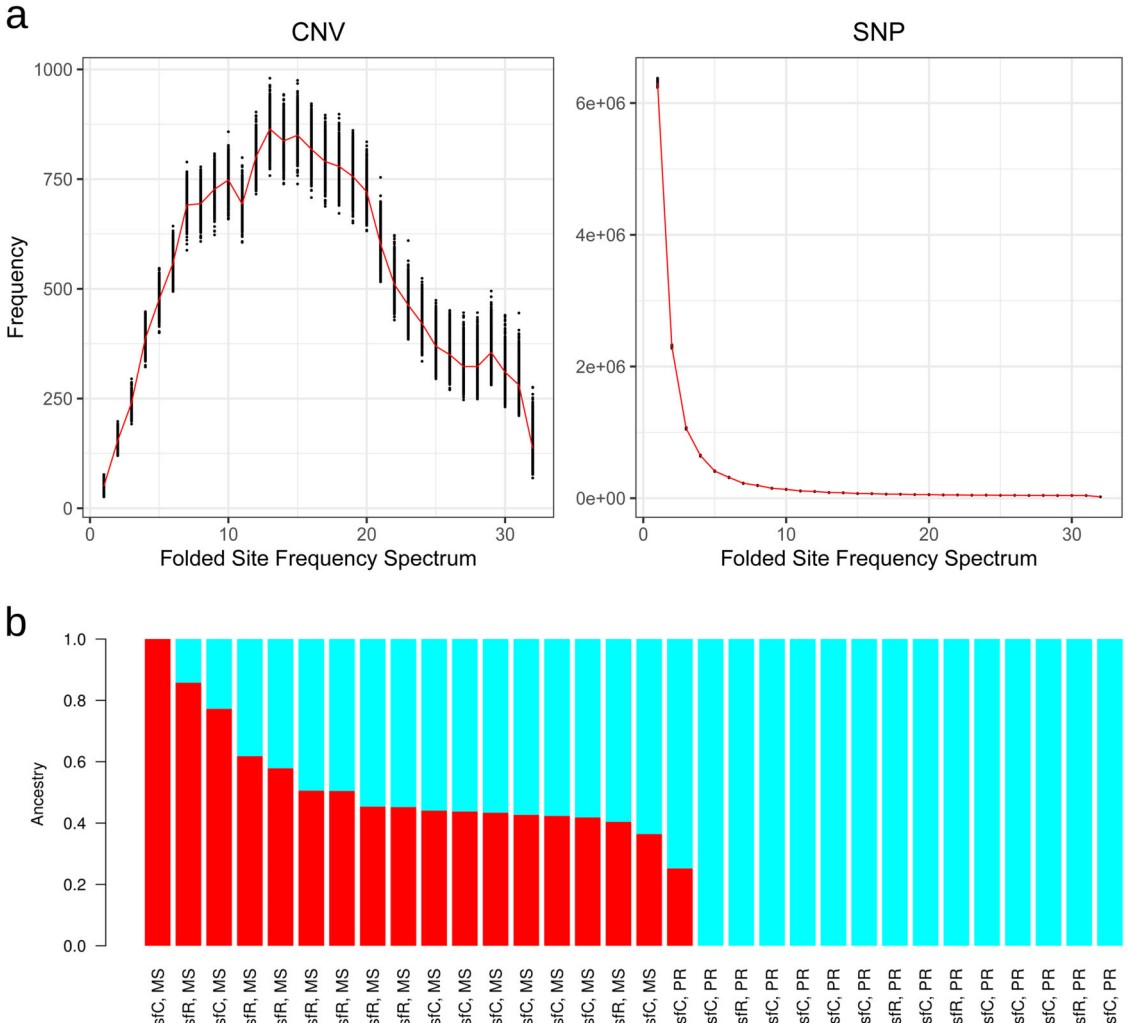

**Fig. 4 Selection on CNVs observed in PR. a** Folded site frequency spectrum at CNV and SNP. The red lines indicate the number of variants with each minor allele frequency, and the black points show the results generated by non-parametric bootstrapping resampled from 100 kb non-overlapping windows with 1000 replications. **b** Results of ancestry coefficient analysis.

CYP6G1 caused DDT-resistance[50]. More specifically, the causative CYP genes may include CYP9A genes, whose CNV was targeted by PR-specific positive selection.

Taken together, these results show that genes playing a key role in insecticide resistance are different from those in the adaptation to host-plants. According to the "pre-adaptation hypothesis", polyphagous insects have a greater ability to cope with toxins from a wide range of host-plants than monophagous ones. Thus, polyphagous insects have a higher potential to generate insecticide resistance[51,52] (but see ref. [53]). In this explanation, the emergence of insecticide resistance is no more than the re-use of existing detoxification genes that are adaptively evolved to host-plants. In polyphagous spider mites, genes that experienced host-plant adaptive evolution contribute to the increased level of insecticide resistance[54], supporting this hypothesis. Our study proposes that even though detoxification machinery genes might largely overlap between host-plant and insecticide resistance, the key player for insecticide resistance can evolve independently from the interaction with host-plants. In other words, the key player genes responsible for the insecticide resistance can be different from those for adaptation to host-plants.

We showed that the genomic distribution of CNVs is generally in line with beneficial effects, implying prevalent positive selection on CNVs. Together with the result showing that detoxification

genes are overrepresented in the genes with CNVs, these CNVs might increase the level of detoxification in a collective way. However, we wish to stress that not all of the beneficial CNVs will be fixed in a population. The probability of fixation of a beneficial mutation is $2s$, where $s$ is the selection coefficient of a beneficial allele. Thus, a very large proportion of mildly beneficial mutations will eventually be lost by genetic drift. In addition, the Hill-Robertson interference effect may hamper the fixation of adaptive CNV as well[55]. The increase of one adaptive CNV of detoxification genes may interfere with that of another adaptive CNV if these two CNVs are genetically linked within a chromosome. Therefore, among the CNVs we observed, those with a strong beneficial effect will be fixed in a population. For this reason, we argue that the list of CNVs under positive selection should not be interpreted as positively *selected* CNVs and that this gene list does not show the direction of adaptive evolution, with the exception of the identified CNV loci with almost complete differentiation ($F_{ST} > 0.8$).

The multiple inter-continental invasions of the fall armyworm are a serious global issue in agriculture. A marker-based study shows that these invasive populations originated from the Florida-Caribbean region[56]. Therefore, it is plausible that the invasive populations may carry the insecticide resistance CNVs identified in this study. The presence of insecticide resistance CNV might explain the success of the first *S. frugiperda* invasion

in West Africa. Assessing the existence of insecticide resistance CNVs in the invasive population and comparative studies on insecticide resistance are urgently needed to test this possibility.

Using genome-scanning, we show that positive selection on the CNV of detoxification genes is related to insecticide resistance in *S. frugiperda* from PR compared with MS. This in silico approach can be complemented by functional genetics studies to verify the role of each positively selected loci CNV in the increased level of insecticide resistance. For example, CRISPR/Cas9[57] or RNAi[58] for each CNVs can be used for this purpose. Importantly, the association between CNV and gene expression level can be useful to verify the role of CNV in the differential levels of insecticide resistance. All these approaches enable us to obtain a more comprehensive and accurate knowledge of insecticide resistance, which will ultimately lead to establishing realistic and effective ways of insect pest control.

## Methods

**Reference genome assembly**. High molecular weight DNA was extracted from one pupa of sfC, which has been raised at our insectarium, using the MagAttract© HMW kit (Qiagen). Single-Molecule-Real Time sequencing was performed using a PacBio RSII (Pacific Biosciences) with 12 SMRT cells (P6-C4 chemistry) at the Get-PlaGe genomic platform (Toulouse, France, https://get.genotoul.fr/). The total throughput was 11,017,798,575 bp in 1,513,346 reads, and the average read length was 7,280 bp. First, we generated assemblies from 166X coverage Illumina paired-end sequences[10] using platanus[59]. Errors in PacBio reads were corrected by this Illumina assembly using Ectools[60], and all uncorrected reads were discarded. The total length of the remaining reads was 11,005,855,683 bp. The error-corrected reads were used to assemble genome sequences using SMARTdenovo[61], and we used pilon[62] with the Illumina paired-end reads to polish the genome assembly. llumina paired-end reads and mate-pair reads[10] were mapped the genome assemblies using bwa[63], and scaffolding was performed using BESST[64].

Pooled high molecular weight genomic DNA was extracted from 10 pupae of sfC, reared at the same insectarium as above. Dovetail™ Hi-C library was constructed from this strain, followed by Illumina Hiseq-X sequencing. Hi-Rise™ software[23] was used to perform scaffolding from the PacBio genome assembly. The correctness of the genome assembly was assessed using BUSCO[26]. The gene annotation was performed using maker[65].

**Resequencing data**. The paired-end resequencing data from *S. frugiperda* samples from Mississippi (MS) was obtained from NCBI SRA (PRJNA494340), which was generated for the fall armyworm genome project[10] and a speciation study[66]. For the *S. frugiperda* sample from Puerto Rico (PR), we collected larvae samples at Santa Isabel (Puerto Rico, see[42] for more detail). The larvae were raised to adults, and gDNA was extracted from thorax using the Promega Wizard® Genomic DNA Purification Kit. Then, 150 bp pair-end Hiseq-4000 sequencing was performed from each individual. Resequencing data had approximately 28.9X and 23.4X coverage per individual for MS and PR, respectively. Adapter sequences were removed using adapterremoval[67]. To identify host strains, we mapped the Illumina reads against mitochondrial genomes (NCBI: KM362176) using bowtie2[68], followed by performing variant calling together with individuals from MS using samtools[69]. The host strain of individuals from MS was already identified from mitochondrial COX1 markers in our previous study[10]. We performed the PCA to observe the grouping according to host strain using picard[70]. For nuclear mapping, the reads were mapped against the reference genome using bowtie2[68] with the -very-sensitive-local preset option. Potential optical or PCR duplicates were removed using Picard tools[70].

**Variant identification**. Variant calling was performed using the GATK-4.0.11.0 package[28]. First, we performed haplotype calling, and the resulting gvcf files were merged into a single gvcf file using the same GATK package. Variants were then called, and only SNPs were extracted among all variants. The number of identified SNPs was 66,529,611. The SNPs were filtered out if their QualByDepth score was less than 2, if FisherStrand score was higher than 60, RMSMappingQuality was less than 40, MappingQualityRankSumTest score was less than $-12.5$, or Read-PosRankSumTest was less than $-8$. The number of remaining SNPs was 16,341,783.

CNVs were inferred from the bam files, which were generated by the mapping of Illumina reads against the reference genome assembly, using CNVcaller[29] with a 100 bp window size. We filtered out CNVs if the minor allele frequency was less than 0.1 to reduce false positives, based on the assumption that the probability of having false positives at the same genomic locus from multiple samples is very low. In other words, we analyzed the CNVs only if CNV was observed from at least four individuals out of 32 (corresponding to allele frequency equal to $4/32 = 0.125$) within 100 bp intervals.

**Population genetics analysis**. The PCA was performed using picard[70] after converting the vcf file to plink format using vcftools 0.1.13[71] and Picard 1.9[70]. Pairwise Weir and Cockerham's $F_{ST}$[72] was calculated using VCFtools[71]. The gene ontology analysis was performed using BinGO 3.0.3[73]. The ancestry coefficient analysis was performed using admixture 1.3.0[74].

**Toxicological bioassay**. Fall armyworm strains used for bioassays included the Benzon strain (Benzon Research Inc, Carlisle, PA) originated from field collections in Mississippi (USA), and the 456LSD4 strain derived from strain 456, which originated from isofamilies of moths collected in Puerto Rico[42]. Derivation of 456LSD4 was through selection of 456 larvae with corn leaf tissue of event TC1507 (producing Cry1F) or 4.75 µg/cm² of Cry1F protoxin on the surface of diet for 5 days. All insect rearing and bioassays were performed in incubators at $26 \pm 2$ °C, 44% relative humidity, and an 18-h light/6-h dark photoperiod.

Commercial grade deltamethrin (Black Flag Extreme Insect Control, 0.32% active ingredient) was from United Industries Corporation (St. Louis, MO, USA), and piperonyl butoxide (Exponent insecticide synergist, 91.3% technical grade) was from MGK (Minneapolis, MN, USA). All pesticides were diluted in distilled water.

The insecticidal activity of deltamethrin alone or in the presence of 16 µl/L of piperonyl butoxide (PBO) was tested against neonates of the Benzon and 456LSD4 strains using bioassays as described elsewhere[75]. Briefly, 8 to 14 doses for each treatment were prepared and used to cover the surface of meridic diet (Beet Armyworm diet, Frontier Agriculture Sciences) in wells of a 128-well bioassay tray (Frontier Agriculture Sciences). We poured 1.5 mL of meridic diet in each well to achieve a 2 cm² diet surface. A 75 µL solution of insecticide or control sample was overlaid on the top of the diet in each well and left to air dry before adding a single neonate in each well. Control treatments included distilled water and resulted in <5% mortality in all bioassay replicates. After treated diet air-drying, one neonate was transferred to each well, and sealed trays were maintained in incubators at 27 °C with a 16:8 (light:dark) photoperiod and 60–80% relative humidity. Bioassays were replicated four (deltamethrin) or two (deltamethrin + PBO) times, with each toxin concentration tested with 16 insects per bioassay replicate. Mortality was scored after 7 days and used for probit analysis in the POLO-PLUS software package[76] to estimate concentrations killing 50% of the individuals (LC₅₀) and corresponding 95% confidence intervals. Significance was determined from non-overlapping 95% confidence intervals. Resistance (lethal dose) ratios and corresponding 95% confidence intervals were estimated from LC₅₀ values in POLO-PLUS. Differences in slopes and intercepts between probit regression lines were examined using a log-likelihood ratio test from chi-square statistics, calculated by POLO-PLUS, and the degree of freedom.

**Statistics and Reproducibility**. This study is based on 15 individuals from PR, and 17 individuals from MS. We assumed that these 32 individuals are genetically unrelated. All statistical analyses from these samples were performed by executing bash, perl, and R scripts. Therefore, all the results can be reproduced by re-executing these scripts. Toxicological bioassays were performed with six independent replicates (four with deltamethrin and two with deltamethrin + PBO). The lethality was calculated from 16 individuals for each toxin concentration for each replicate.

**Reporting summary**. Further information on research design is available in the Nature Research Reporting Summary linked to this article.

## Data availability

The reference genome assembly generated in this study is available at NCBI (JACWZW000000000), together with raw PacBio reads and Hi-C data (PRJNA662887). The reference genome assembly is also available at BIPAA (www.bipaa.genouest.org/sp/spodoptera_frugiperda). The resequencing data in this study are available at NCBI SRA (PRJNA494340, PRJNA577869). The vcf files generated in this study are available at Zenodo[77]. Raw data used to generate figures are available as source data files within the Supplementary Materials, together with R scripts used to generate corresponding figures. Raw data used to generate Supplementary figures are also available as a source data file within the Supplementary Materials.

## Code availability

All custom perl and R scripts used in this study are available at Zenodo[78].

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

## Acknowledgements

This work was supported by a grant from the department of Santé des Plantes et Environnement at Institut national de la recherche agronomique (adaptivesv). Partial support for this project was also provided by Agriculture and Food Research Initiative Foundational Program competitive grant No. 2018-67013-27820 from the USDA National Institute of Food and Agriculture. We also acknowledge Yutao Xiao (CAAS) for discussions on this paper and Claire Lemaitre (INRIA) for CNV identification.

## Author contributions

K.N.: Conceptualization; Formal Analysis; Funding Acquisition, Investigation, Project Administration, Writing the paper; S.G.: Preparation of data for sequencing; F.H.: Manual gene annotation; C.A.B., S.H.: Acquisition of samples for sequencing; A.B., F.L.: Generation of the reference genome, automatic gene annotation; N.N., E.A.: The acquisition of Illumina sequencing data from the population in Mississippi; H.A., J.L.J-F: Design, performance, and analysis of toxicological bioassays; G.L.G.: Design of toxicological bioassays. All co-authors participated in the writing of this paper, and they approved it.

## Competing interests

The authors declare no competing interests.
