## [Peer Review File · Communications Biology]

Reviewers' Comments:

Reviewer #1:

Remarks to the Author:

Nam et al. report on CNV between two geographic populations with different insecticide resistance levels. They found that a locus with high allelic differentiation contained a group of detoxification genes and conclude that CNV of detoxification (P450) genes is responsible for insecticide resistance. In my opinion, the manuscript has several drawbacks and, in the current state, cannot be considered for publication in *Communications in Biology*

-At the end of the abstract authors state the following: "CNV of detoxification genes is responsible for insecticide resistance and occurred independently from host plant". This conclusion is very bold as they only included two different geographical populations with different resistance levels in their study. Author should add more populations from different geographic regions if they would like to include such general statements.

-line 159/160: the authors found CNV of P450 genes and alcohol dehydrogenase genes and then link the role of these P450s with insecticide resistance using P450 expression data from other *Spodoptera* species, without providing any functional data (functional expression, enzyme activity, synergism data, CRISPR?). I think at least some functional data should be added before any statement regarding the role of P450 CNV in insecticide resistance levels of *S. frugiperda* can be made

-last, and most importantly, the authors mention that there is difference in resistance levels between the MS and PR population (line 67-69). However, as the authors did not include insecticide toxicity data for the sequenced MS or PR individuals, one cannot know whether there is any difference in insecticide resistance for the populations of this study and, hence, cannot also make any statement regarding the possible role of P450 CNV in insecticide resistance.

Minor comment:

-PRJNA494340 is not yet publicly available.

-line In M&M the authors mention that there is 20X Coverage per PR individual. Is the coverage similar for MS individuals? Could a difference in coverage influence CNV analysis? Authors should provide add more details regarding the MS resequencing.

Reviewer #2:

Remarks to the Author:

Nice work for CNVs in *Spodoptera frugiperda*, very well written with critical discussion

Reviewer #3:

Remarks to the Author:

The manuscript COMMSBIO-20-0108-T, "Adaptation by copy number variation increases insecticide resistance in fall armyworms" reports the use of genomic copy number variations between two host strains of *Spodoptera frugiperda* in relation to insecticide resistance (based on geographic origination) as a means to explain if host-plant adaptation resulting in multi-copy detoxification genes explains insecticide resistance. Whereas they detected significant allelic differentiation of genomic copy number variation between two geographic populations, this differentiation was not observed between host strains. Their conclusion based on these findings was that copy number variation of detoxification genes is responsible for insecticide resistance, but

that the emergence of insecticide resistances occurred independently from host-plant adaptation.

Overall the authors did a very good job of laying out the methodology, collecting and presenting the pertinent data, and testing their hypothesis. The results presented are unambiguous and the results stand for themselves.

I have several minor points of concern.

Small sample size. The investigators evaluated 9 sfC and 8 sfR from Mississippi, and 11 sfC and 4 sfR from Puerto Rico. Evaluating only 4 sfR from Puerto Rico is not much of a sample. Granted the copy number variation shows a distinct grouping but there is one sfR from Puerto Rico that stands out from the other CNV for that geographic locale. Alone, this would give greater concern, but with the support of the SNP principle component analysis showing a tight grouping among sfR strains from Puerto Rico I think it is acceptable.

Another point of concern is with Lines 181-182. Here the authors proposed the possibility that increases in the chitinase gene copy may be associated with Bt-resistance in Puerto Rico. This is not likely. Banjerjee et al. 2017 identified the mechanism of Cry1F resistance in *Spodoptera frugiperda* in Puerto Rico to be linked to a mutation in ATP binding cassette C2 (ABCC2) gene that functions as the Cry1Fa receptor. Since the mechanism of resistance is known, there is no need to speculate otherwise.

Another point of concern. The authors make the assumption that CNV in detoxification genes is responsible for insecticide resistance, but did not demonstrate that the Puerto Rico strains used in this study were in fact resistant to any insecticides. Nor do they know if the Mississippi strains lack resistance to insecticides. They do elude to this in line 285-298. I do not think this lack of substantiation diminishes the relevance of the manuscript, but the authors should not assume the insecticide resistance exists but should state that the CNV of detoxification genes "may" result in increased incidence in insecticide resistance.

I think there may be more genetic exchange between Caribbean/Florida *Spodoptera frugiperda* and some other populations present in the southern U.S. during a given year. *S. frugiperda* does not overwinter in the southern U.S. except in Florida and southern Texas. Thus, populations collected in Mississippi are migratory. The Appalachian Mountains are not a barrier between Florida, Georgia and westward (the range stops in northern Georgia) and *S. frugiperda* moths can easily move upon tropical storms from the Caribbean into the southern U.S. Huang et al. 2014 reported high Cry1F resistance allele frequency from populations collected in Louisiana. These alleles likely originated in the Caribbean or Florida. Perhaps your ability to genetically separate geographic populations might differ depending in the migratory movement during a given year.

Other remarks are simply grammatical.

Line 18. Strike "main"

Line 46. Do not capitalize Tobacco

Line 50. Reword ..detoxification genes themselves are not necessarily...

Line 60. Strike "all"

Line 70. Strike "massive" and insert frequent

Line 145-146. Reword. The Z chromosomes in female were paternally inherited, whereas the mitochondrial genomes were maternally inherited.

Line 166. Do not capitalize Beet and Tobacco

Line 244-245. Reword. From these observations, we conclude that CNV of detoxification genes is probably responsible for increased insecticide resistance in Puerto Rico.

Reviewer 1

Reviewer #1 (Remarks to the Author):

Nam et al. report on CNV between two geographic populations with different insecticide resistance levels. They found that a locus with high allelic differentiation contained a group of detoxification genes and conclude that CNV of detoxification (P450) genes is responsible for insecticide resistance. In my opinion, the manuscript has several drawbacks and, in the current state, cannot be considered for publication in *Communications in Biology*

Thank you very much for your comments. We found that all comments from you were very helpful. By answering your comments, we believe that the manuscript is improved.

-At the end of the abstract authors state the following: "CNV of detoxification genes is responsible for insecticide resistance and occurred independently from host plant". This conclusion is very bold as they only included two different geographical populations with different resistance levels in their study. Author should add more populations from different geographic regions if they would like to include such general statements.

R1-1: The last sentence we wrote in the initially submitted manuscript is, "From this result, we concluded in fall armyworm that the copy number variation of detoxification genes is responsible for insecticide resistance and that the emergence of insecticide resistance occurred independently from host-plant adaptation," where the occurrence of CNVs is not mentioned.

We assume that you did not mean the occurrence of CNVs. It is true that we need to analyze a very large number of geographic populations to show population-specific occurrences of CNVs. The occurrence of CNV is beyond the scope of this study, and we do not believe that CNV occurs independently from host-plant because the occurrence of CNV is no more than a mutation process.

We now acknowledge that the sentence could be read as a general conclusion in the fall armyworm. We modified the sentence to make the meaning clear and not to mention that the conclusion is not about 'the' fall armyworm (Line32-L34 at abstract and L270 at Discussion).

-line 159/160: the authors found CNV of P450 genes and alcohol dehydrogenase genes and then link the role of these P450s with insecticide resistance using P450 expression data from other *Spodoptera* species, without providing any functional data (functional expression, enzyme activity, synergism data, CRISPR?). I think at least some functional data should be added before any statement regarding the role of P450 CNV in insecticide resistance levels of *S. frugiperda* can be made

R1-2: We completely agree with you at this point. To address your concern, we performed a comparative toxicological bioassay between a lab strain from Puerto Rico (456LSD4, which was seeded from the PR population used in this study) and another lab strain from Mississippi (Benzon).

When these two strains were treated by deltamethrin, 456LSD4 has a higher resistance factor than Benzon by 6.5 times at LC_{50} . A log-likelihood ratio test shows Benzon has a steeper slope of probit regression lines between the dose of deltamethrin and the proportion of killed larvae ($p = 6.16 \times 10^{-7}$), supporting that 456LSD4 has a stronger

resistance than Benzon. This result provides experimental supports that the Puerto Rico population has stronger insecticide resistance than the Mississippi population.

To investigate the functional role of P450 in insecticide resistance, we treated the 456LSD4 resistant strain with PBO (Piperonyl butoxide), a well-known inhibitor of P450 enzymes. The resistance was abolished by this treatment. Indeed, 456LSD4 strain has overlapping 95% confidence intervals with Benzon at LC_{50} (0.19-0.31 and 0.22-0.38 for Benzon and 456LSD4, respectively). Log-likelihood ratio test shows that probit linear regression lines do not have significantly different slopes between Benzon and PBO-treated 456LSD4 ($p = 0.215$). This result demonstrates that the observed differential resistance between Mississippi and Puerto Rico population is due to P450.

These new results can be found at L31-L32 (abstract), L82-L84 (introduction), L195-L223 (results), L280-L292 (discussion), L401-L430 (methods), and table 2.

-last, and most importantly, the authors mention that there is difference in resistance levels between the MS and PR population (line 67-69). However, as the authors did not include insecticide toxicity data for the sequenced MS or PR individuals, one cannot know whether there is any difference in insecticide resistance for the populations of this study and, hence, cannot also make any statement regarding the possible role of P450 CNV in insecticide resistance.

R1-3: Please see the answer above (R1-2)

Minor comment:

-PRJNA494340 is not yet publicly available.

R1-4: This data was generated by another study, which is under the publication process at another journal. The data is embargoed yet. However, upon the acceptance of this paper by *Communications Biology*, we will release this data immediately.

-line In M&M the authors mention that there is 20X Coverage per PR individual. Is the coverage similar for MS individuals? Could a difference in coverage influence CNV analysis? Authors should provide add more details regarding the MS resequencing.

R1-5: We attempted to provide more information in the result section (L367-L368).

Reviewer 2

Reviewer #2 (Remarks to the Author):

Nice work for CNVs in *Spodoptera frugiperda*, very well written with critical discussion

R2-1: Thank you very much for the positive review.

Reviewer 3

Reviewer #3 (Remarks to the Author):

The manuscript COMMSBIO-20-0108-T, "Adaptation by copy number variation increases insecticide resistance in fall armyworms" reports the use of genomic copy number variations between two host strains of *Spodoptera frugiperda* in relation to insecticide resistance (based on geographic origination) as a means to explain if host-plant adaption resulting in multi-copy detoxification genes explains insecticide resistance. Whereas they detected significant allelic differentiation of genomic copy number variation between two geographic populations, this differentiation was not observed between host strains. Their

conclusion based on these findings was that copy number variation of detoxification genes is responsible for insecticide resistance, but that the emergence of insecticide resistances occurred independently from host-plant adaptation.

Overall the authors did a very good job of laying out the methodology, collecting and presenting the pertinent data, and testing their hypothesis. The results presented are unambiguous and the results stand for themselves.

Thank you very much for your comments. We found that all the comments from you are helpful. By revising the manuscript according to your comments, we believe that the manuscript is now improved.

I have several minor points of concern.

Small sample size. The investigators evaluated 9 sfC and 8 sfR from Mississippi, and 11 sfC and 4 sfR from Puerto Rico. Evaluating only 4 sfR from Puerto Rico is not much of a sample. Granted the copy number variation shows a distinct grouping but there is one sfR from Puerto Rico that stands out from the other CNV for that geographic locale. Alone, this would give greater concern, but with the support of the SNP principle component analysis showing a tight grouping among sfR strains from Puerto Rico I think it is acceptable.

R3-1: We acknowledge that four samples from sfR are not very high. According to population genetics theory, time to coalescence is proportional to the effective population size. As the population size of the fall armyworm is expected to be very high (like millions), the observed genetic variants are expected to be generated anciently. Therefore, the observed genomic variants from the fall armyworm have information of long evolutionary time. As a consequence, population genomic studies in a species with a large population size do not require a very large number of samples, and we believe that four samples should provide sufficient informative genomic variants.

Regarding the CNV pattern of principal component analysis, we agree with you that one outlier from PR is not a concern because SNVs show a clear grouping among PR individuals.

Another point of concern is with Lines 181-182. Here the authors proposed the possibility that increases in the chitinase gene copy may be associated with Bt-resistance in Puerto Rico. This is not likely. Banerjee et al. 2017 identified the mechanism of Cry1F resistance in *Spodoptera frugiperda* in Puerto Rico to be linked to a mutation in ATP binding cassette C2 (ABCC2) gene that functions as the Cry1Fa receptor. Since the mechanism of resistance is known, there is no need to speculate otherwise.

R3-2: We agree with you in that other studies already identified mutations on ABCC2 genes as causal genetic variants for the Bt-resistance. These studies, however, do not argue that they identified all the causal genetic variants of field-evolved Bt-resistance. Therefore, we wish to open the possibility that other genetic variants might contribute to Bt-resistance as well. We revised the sentence to mention to make this point clear (Line183-L186).

Another point of concern. The authors make the assumption that CNV in detoxification genes is responsible for insecticide resistance, but did not demonstrate that the Puerto Rico strains used in this study were in fact resistant to any insecticides. Nor do they know if the Mississippi strains lack resistance to insecticides. They do elude to this in line 285-298. I do not think this lack of substantiation diminishes the relevance of the manuscript, but the authors should not assume the insecticide resistance exists but should state that the CNV of detoxification genes “may” result in increased incidence in insecticide

resistance.

R3-3: We agree with you completely. We performed comparative toxicological bioassays using deltamethrin to show that the Puerto Rico population has a higher resistance, and the increased resistance is due to P450 genes.

We observed that 456LSD4 (a lab strain seed from Puerto Rico's population used in this study) has higher LC_{50} than Benzon (a lab strain originated from Mississippi), showing that 456LSD4 has a stronger resistance than Benzon. When the resistant strain 456LSD4 was treated by piperonyl butoxide, a well-known inhibitor of P450 enzymes, the difference between 456LSD4 and Benzon disappears. This result provides experimental evidence that P450 plays a key role in the deltamethrin resistance in Puerto Rico's population. These new results can be found at L31-L32 (abstract), L82-L84 (introduction), L195-L223 (results), L280-L292 (discussion), L401-L430 (methods), and table 2.

I think there may be more genetic exchange between Caribbean/Florida *Spodoptera frugiperda* and some other populations present in the southern U.S. during a given year. *S. frugiperda* does not overwinter in the southern U.S. except in Florida and southern Texas. Thus, populations collected in Mississippi are migratory. The Appalachian Mountains are not a barrier between Florida, Georgia and westward (the range stops in northern Georgia) and *S. frugiperda* moths can easily move upon tropical storms from the Caribbean into the southern U.S. Huang et al. 2014 reported high Cry1F resistance allele frequency from populations collected in Louisiana. These alleles likely originated in the Caribbean or Florida. Perhaps your ability to genetically separate geographic populations might differ depending in the migratory movement during a given year.

R3-4: You are absolutely right. As our experimental bioassay demonstrated that Puerto Rico's population indeed has a higher resistance than Mississippi's population, we do not believe that the corresponding paragraph is relevant anymore. Therefore, we removed this paragraph.

Other remarks are simply grammatical.

Line 18. Strike "main"

Line 46. Do not capitalize Tobacco

Line 50. Reword ..detoxification genes themselves are not necessarily...

Line 60. Strike "all"

Line 70. Strike "massive" and insert frequent

Line 145-146. Reword. The Z chromosomes in female were paternally inherited, whereas the mitochondrial genomes were maternally inherited.

Line 166. Do not capitalize Beet and Tobacco

Line 244-245. Reword. From these observations, we conclude that CNV of detoxification genes is probably responsible for increased insecticide resistance in Puerto Rico.

R3-5: We revised the manuscript according your suggestions. Thank you for your thorough revision.

Reviewers' Comments:

Reviewer #1:

Remarks to the Author:

The authors have addressed most of my comments in an adequate way. The quality of the manuscript is now sufficient for publication in *Comms. Biology*.

Minor comment:

Please soften the tone of the last sentence of the abstract into:

"Our results indicate that copy number variation of detoxification genes might be responsible for insecticide resistance in fall armyworm and that evolutionary forces causing insecticide resistance could be independent of host-plant adaptation."

Reviewer #3:

Remarks to the Author:

The authors adequately addressed all of my concerns.

Couple of minor comments.

Line 171 refers to *H. armigera*. Since is the first reference to this species spell out *Helicoverpa*. Then on Line 172 it is spelled out and *H. armigera* can be used here.

Line 417. How much meridic diet and insecticide solution was used in the overlays?

REVIEWERS' COMMENTS:

Reviewer #1 (Remarks to the Author):

The authors have addressed most of my comments in an adequate way. The quality of the manuscript is now sufficient for publication in Comms. Biology.

Minor comment:

Please soften the tone of the last sentence of the abstract into:

"Our results indicate that copy number variation of detoxification genes might be responsible for insecticide resistance in fall armyworm and that evolutionary forces causing insecticide resistance could be independent of host-plant adaptation."

R1: We changed the sentence according to your suggestion (Line34-Line36).

Reviewer #3 (Remarks to the Author):

The authors adequately addressed all of my concerns.

Couple of minor comments.

Line 171 refers to *H. armigera*. Since is the first reference to this species spell out *Helicoverpa*. Then on Line 172 it is spelled out and *H. armigera* can be used here.

R3-1: We corrected these words as you indicated (Line172-L173).

Line 417. How much meridic diet and insecticide solution was used in the overlays?

R3-2: We poured 1.5 mL of meridic diet in each well to achieve a 2 cm² diet surface. A 75 µL solution of insecticide or control sample was overlaid on the top of the diet in each well and left to air dry before adding a single neonate in each well. We inserted this description in the manuscript (L419-L421).